# Exploiting the Structure:
# Stochastic Gradient Methods Using Raw Clusters[*]

**Zeyuan Allen-Zhu**[†]
Princeton University / IAS
zeyuan@csail.mit.edu

**Yang Yuan**[†]
Cornell University
yangyuan@cs.cornell.edu

**Karthik Sridharan**
Cornell University
sridharan@cs.cornell.edu

## Abstract

The amount of data available in the world is growing faster than our ability to deal with it. However, if we take advantage of the internal *structure*, data may become much smaller for machine learning purposes. In this paper we focus on one of the fundamental machine learning tasks, empirical risk minimization (ERM), and provide faster algorithms with the help from the clustering structure of the data.

We introduce a simple notion of *raw clustering* that can be efficiently computed from the data, and propose two algorithms based on clustering information. Our accelerated algorithm ClusterACDM is built on a novel Haar transformation applied to the dual space of the ERM problem, and our variance-reduction based algorithm ClusterSVRG introduces a new gradient estimator using clustering. Our algorithms outperform their classical counterparts ACDM and SVRG respectively.

## 1 Introduction

For large-scale machine learning applications, $n$, the number of training data examples, is usually very large. To search for the optimal solution, it is often desirable to use *stochastic* gradient methods which only require one (or a batch of) random example(s) from the given training set per iteration in order to form an *estimator* of the true gradient.

For empirical risk minimization problems (ERM) in particular, stochastic gradient methods have received a lot of attention in the past decade. The original stochastic gradient descent (SGD) [4, 26] simply defines the estimator using one random data example and converges slowly. Recently, variance-reduction methods were introduced to improve the running time of SGD [6, 7, 13, 18, 20–22, 24], and accelerated gradient methods were introduced to further improve the running time when the regularization parameter is small [9, 16, 17, 23, 27].

None of the above cited results, however, have considered the *internal structure* of the dataset, that is, using the stochastic gradient with respect to one data vector $p$ to estimate the stochastic gradients of other data vectors close to $p$. To illustrate why internal structure can be helpful, consider the following extreme case: if all the data vectors are located at the same spot, then every stochastic gradient represents the full gradient of the entire dataset. In a non-extreme case, if data vectors form *clusters*, then the stochastic gradient of one data vector could provide a rough estimation for its neighbors. Therefore, one should expect ERM problems to be *easier* if the data vectors are clustered.

More importantly, well-clustered datasets are *abundant* in big-data scenarios. For instance, although there are more than 1 billion of users on Facebook, the intrinsic "feature vectors" of these users can be naturally categorized by the users' occupations, nationalities, etc. As another example, although there

---

[*]The full version of this paper can be found on https://arxiv.org/abs/1602.02151.

[†]These two authors equally contribute to this paper.

are 581,012 vectors in the famous Covtype dataset [8], these vectors can be efficiently categorized into 1,445 clusters of diameter $0.1$ — see Section 5. With these examples in mind, we investigate in this paper how to train an ERM problem faster using clustering information.

## 1.1 Known Result and Our Notion of Raw Clustering

In a seminal work by Hofmann et al. published in NIPS 2015 [11], they introduced N-SAGA, the first ERM training algorithm that takes into account the similarities between data vectors. In each iteration, N-SAGA computes the stochastic gradient of one data vector $p$, and uses this information as a biased representative for a small neighborhood of $p$ (say, 20 nearest neighbors of $p$).

In this paper, we focus on a more general and powerful notion of clustering yet capturing only the *minimum* requirement for a cluster to have similar vectors. Assume without loss of generality data vectors have norm at most 1. We say that a partition of the data vectors is an $(s, \delta)$ *raw clustering* if the vectors are divided into $s$ disjoint sets, and the *average* distance between vectors in each set is at most $\delta$. For different values of $\delta$, one can obtain an $(s_\delta, \delta)$ raw clustering where $s_\delta$ is a function on $\delta$. For example, a $(1445, 0.1)$ raw clustering exists for the Covtype dataset that contains 581,012 data vectors. Raw clustering enjoys the following nice properties.

- It allows outliers to exist in a cluster and nearby vectors to be split into multiple clusters.
- It allows large clusters. This is in contrast to N-SAGA which requires each cluster to be very small (say of size 20) due to their algorithmic limitation.

**Computation Overhead.** Since we do not need exactness, raw clusterings can be obtained very efficiently. We directly adopt the approach of Hofmann et al. [11] because finding approximate clustering is the same as finding approximate neighbors. Hofmann et al. [11] proposed to use approximate nearest neighbor algorithms such as LSH [2, 3] and product quantization [10, 12], and we use LSH in our experiments. Without trying hard to optimize the code, we observed that in time $0.3T$ we can detect if good clustering exists, and if so, in time around $3T$ we find the actual clustering. Here $T$ is the running time for a stochastic method such as SAGA to perform $n$ iterations (i.e., one pass) on the dataset.

We repeat three remarks from Hofmann et al. First, the better the clustering quality the better performance we can expect; yet one can always use the trivial clustering as a fallback option. Second, the clustering time should be amortized over multiple runs of the training program: if one performs 30 runs to choose between loss functions and tune parameters, the amortized cost to compute a raw clustering is at most $0.1T$. Third, since stochastic gradient methods are sequential methods, increasing the computational cost in a highly parallelizable way may not affect data throughput.

NOTE. Clustering can also be obtained for free in some scenarios. If Facebook data are retrieved, one can use the geographic information of the users to form raw clustering. If one works with the CIFAR-10 dataset, the known CIFAR-100 labels can be used as clustering information too [14].

## 1.2 Our New Results

We first observe some limitations of N-SAGA. Firstly, it is *biased* algorithm and does not converge to the objective minimum.[3] Secondly, in order to keep the bias small, N-SAGA only exploits a small neighborhood for every data vector. Thirdly, N-SAGA may need 20 times more computation time per iteration as compared to SAGA or SGD, if 20 is the average neighborhood size.

We explore in this paper how a given $(s, \delta)$ raw clustering can improve the performance of training ERM problems. We propose two unbiased algorithms that we call ClusterACDM and ClusterSVRG. The two algorithms use different techniques. ClusterACDM uses a novel clustering-based transformation in the dual space, and provides a faster algorithm than ACDM [1, 15] both in practice and in terms of asymptotic worst-case performance. ClusterSVRG is built on top of SVRG [13], but using a new clustering-based gradient estimator to improve the running time.

More specifically, consider *for simplicity* ridge regression where the $\ell_2$ regularizer has weight $\lambda > 0$. The best known non-accelerated methods (such as SAGA [6] and SVRG [6]) and the best known accelerated methods (such as ACDM or AccSDCA [23])) run in time respectively

$$\text{non-accelerated: } \widetilde{O}\left(nd + \frac{d}{\lambda}\right) \quad \text{and} \quad \text{accelerated: } \widetilde{O}\left(nd + \frac{\sqrt{n}}{\sqrt{\lambda}}d\right) \tag{1.1}$$

where $d$ is the dimension of the data vectors and the $\widetilde{O}$ notation hides the $\log(1/\varepsilon)$ factor that depends on the accuracy. Accelerated methods converge faster when $\lambda$ is smaller than $1/n$.

Our ClusterACDM method outperforms (1.1) both in terms of theory and practice. Given an $(s, \delta)$ raw clustering, ClusterACDM enjoys a worst-case running time

$$\widetilde{O}\Big(nd + \tfrac{\max\{\sqrt{s}, \sqrt{\delta n}\}}{\sqrt{\lambda}}d\Big) \ . \tag{1.2}$$

In the ideal case when all the feature vectors are identical, ClusterACDM converges in time $\widetilde{O}\big(nd + \tfrac{d}{\sqrt{\lambda}}\big)$. Otherwise, our running time is asymptotically better than known accelerated methods by a factor $O(\min\{\sqrt{\tfrac{n}{s}}, \tfrac{1}{\sqrt{\delta}}\})$ that depends on the clustering quality. Our speed-up also generalizes to other ERM problems as well such as Lasso.

Our ClusterSVRG matches the best non-accelerated result in (1.1) in the worst-case;[4] however, it enjoys a provably smaller variance than SVRG or SAGA, so runs faster in practice.

**Techniques Behind ClusterACDM.** We highlight our main techniques behind ClusterACDM. Since a cluster of vectors have almost identical directions if $\delta$ is small, we wish to create an auxiliary vector for each cluster representing "moving in the average direction of all vectors in this cluster". Next, we design a stochastic gradient method that, instead of uniformly choosing a random vector, selects those auxiliary vectors with a much higher probability compared with ordinary ones. This could lead to a running time improvement because moving in the direction of an auxiliary vector only costs $O(d)$ running time but exploits the information of the entire cluster.

We implement the above intuition using *optimization* insights. In the dual space of the ERM problem, each variable corresponds to a data example in the primal, and the objective is known to be coordinate-wise smooth with the same smoothness parameter per coordinate. In the preprocessing step, ClusterACDM applies a novel *Haar transformation* on each cluster of the dual coordinates. Haar transformation rotates the dual space, and for each cluster, it automatically reserves a new dual variable that corresponds to the "auxiliary vector" mentioned above. Furthermore, these new dual variables have significantly larger smoothness parameters and therefore will be selected with probability much larger than $1/n$ if one applies a state-of-the-art accelerated coordinate descent method such as ACDM.

**Other Related Work.** ClusterACDM can be viewed as "preconditioning" the data matrix from the dual variable side. Recently, preconditioning received some attentions in machine learning. In particular, non-uniform sampling can be viewed as using diagonal preconditioners [1, 28]. However, diagonal preconditioning has nothing to do with clustering: for instance, if all data vectors have the same Euclidean norm, the cited results are identical to SVRG or APCG so do not exploit the clustering information. Some authors also study preconditioning from the primal side using SVD [25]. This is different from us because for instance when all the data vectors are same (thus forming a perfect cluster), the cited result reduces to SVRG and does not improve the running time.

## 2 Preliminaries

Given a dataset consisting of $n$ vectors $\{a_1, \ldots, a_n\} \subset \mathbb{R}^d$, we assume without loss of generality that $\|a_i\|_2 \le 1$ for each $i \in [n]$. Let a *clustering* of the dataset be a partition of the indices $[n] = S_1 \cup \cdots \cup S_s$. We call each set $S_c$ a *cluster* and use $n_c = |S_c|$ to denote its size. It satisfies $\sum_{c=1}^{s} n_c = n$. We are interested in the following quantification that estimates the clustering quality:

**Definition 2.1** (raw clustering on vectors). *We say a partition $[n] = S_1 \cup \cdots \cup S_s$ is an $(s, \delta)$ raw clustering for the vectors $\{a_1, \ldots, a_n\}$ if for every cluster $S_c$ it satisfies $\frac{1}{|S_c|^2} \sum_{i,j \in S_c} \|a_i - a_j\|^2 \le \delta$.*

We call it a *raw* clustering because the above definition captures the *minimum* requirement for each cluster to have similar vectors. For instance, the above "average" definition allows a few outliers to exist in each cluster and allows nearby vectors to be split into different clusters.

Raw clustering of the dataset is very easy to obtain: we include in Section 5.1 a simple and efficient algorithm for computing an $(s_\delta, \delta)$ raw clustering of any quality $\delta$. A similar assumption like our $(s, \sigma)$ raw clustering assumption in Definition 2.1 was also introduced by Hofmann et al. [11].

**Definition 2.2** (Smoothness and strong convexity). *For a convex function $g \colon \mathbb{R}^n \to \mathbb{R}$,*

- *$g$ is $\sigma$-strongly convex if $\forall x, y \in \mathbb{R}^n$, it satisfies $g(y) \ge g(x) + \langle \nabla g(x), y - x \rangle + \frac{\sigma}{2}\|x - y\|^2$.*
- *$g$ is $L$-smooth if $\forall x, y \in \mathbb{R}^n$, it satisfies $\|\nabla g(x) - \nabla g(y)\| \le L\|x - y\|$.*

- $g$ *is coordinate-wise smooth with parameters* $(L_1, L_2, \ldots, L_n)$, *if for every* $x \in \mathbb{R}^n$, $\delta > 0$, $i \in [n]$, *it satisfies* $|\nabla_i g(x + \delta \mathbf{e}_i) - \nabla_i g(x)| \leq L_i \cdot \delta$.

For strongly convex and coordinate-wise smooth functions $g$, one can apply the accelerated coordinate descent algorithm (ACDM) to minimize $g$:

**Theorem 2.3** (ACDM). *If* $g(x)$ *is* $\sigma$-*strongly convex and coordinate-wise smooth with parameters* $(L_1, \ldots, L_n)$, *the non-uniform accelerated coordinate descent method of [1] produces an output* $y$ *satisfying* $g(y) - \min_x g(x) \leq \varepsilon$ *in* $O\big(\sum_i \sqrt{L_i/\sigma} \cdot \log(1/\varepsilon)\big)$ *iterations. Each iteration runs in time proportional to the computation of a coordinate gradient* $\nabla_i g(\cdot)$ *of* $g$.

**Remark 2.4.** Accelerated coordinate descent admits several variants such as APCG [17], ACDM [15], and NU_ACDM [1]. These variants agree on the running time when $L_1 = \cdots = L_n$, but NU_ACDM is the fastest when $L_1, \ldots, L_n$ are *non-uniform*. More specifically, NU_ACDM selects a coordinate $i$ with probability proportional to $\sqrt{L_i}$. In contrast, ACDM samples coordinate $i$ with probability proportional to $L_i$ and APCG samples $i$ with probability $1/n$. We refer to NU_ACDM as the accelerated coordinate descent method (ACDM) in this paper.

## 3 ClusterACDM Algorithm

Our ClusterACDM method is an *accelerated* stochastic gradient method just like AccSDCA [23], APCG [17], ACDM [1, 15], SPDC [27], etc. Consider a regularized least-square problem

$$\text{Primal:} \quad \min_{x \in \mathbb{R}^d} \left\{ P(x) \stackrel{\text{def}}{=} \frac{1}{2n} \sum_{i=1}^n (\langle a_i, x \rangle - l_i)^2 + r(x) \right\} , \tag{3.1}$$

where each $a_i \in \mathbb{R}^d$ is the feature vector of a training example and $l_i$ is the label of $a_i$. Problem (3.1) becomes *ridge regression* when $r(x) = \frac{\lambda}{2}\|x\|_2^2$, and becomes *Lasso* when $r(x) = \lambda\|x\|_1$. One of the state-of-the-art *accelerated* stochastic gradient methods to solve (3.1) is through its dual. Consider the following equivalent dual formulation of (3.1) (see for instance [17] for the detailed proof):

$$\text{Dual:} \quad \min_{y \in \mathbb{R}^n} \left\{ D(y) \stackrel{\text{def}}{=} \frac{1}{2n}\|y\|^2 + \frac{1}{n}\langle y, l \rangle + r^*\big(-\frac{1}{n}Ay\big) \right.$$
$$\left. = \frac{1}{n}\sum_{i=1}^n \big(\frac{1}{2}y_i^2 + y_i \cdot l_i\big) + r^*\big(-\frac{1}{n}\sum_{i=1}^n y_i a_i\big) \right\} , \tag{3.2}$$

where $A = [a_1, a_2, \ldots, a_n] \in \mathbb{R}^{d \times n}$ and $r^*(y) \stackrel{\text{def}}{=} \max_w y^T w - r(w)$ is the Fenchel dual of $r(w)$.

### 3.1 Previous Solutions

If $r(x)$ is $\lambda$-strongly convex in $P(x)$, the dual objective $D(y)$ is both strongly convex and smooth. The following lemma is due to [17] but is also proved in our appendix for completeness.

**Lemma 3.1.** *If* $r(x)$ *is* $\lambda$-*strongly convex, then* $D(y)$ *is* $\sigma = \frac{1}{n}$ *strongly convex and coordinate-wise smooth with parameters* $(L_1, \ldots, L_n)$ *for* $L_i = \frac{1}{n} + \frac{1}{\lambda n^2}\|a_i\|^2$.

For this reason, the authors of [17] proposed to apply accelerated coordinate descent (such as their APCG method) to minimize $D(y)$.[5] Assuming without loss of generality $\|a_i\|^2 \leq 1$ for $i \in [n]$, we have $L_i \leq \frac{1}{n} + \frac{1}{\lambda n^2}$. Using Theorem 2.3 on $D(\cdot)$, we know that ACDM produces an $\varepsilon$-approximate dual minimizer $y$ in $O(\sum_i \sqrt{L_i/\sigma} \log(1/\varepsilon)) = \widetilde{O}(n + \sqrt{n/\lambda})$ iterations, and each iteration runs in time proportional to the computation of $\nabla_i D(y)$ which is $O(d)$. This total running time $\widetilde{O}(nd + \sqrt{n/\lambda} \cdot d)$ is the fastest for solving (3.1) when $r(x)$ is $\lambda$-strongly convex.

Due to space limitation, in the main body we *only focus* on the case when $r(x)$ is strongly convex; the non-strongly convex case (such as Lasso) can be reduced to this case. See Remark A.1 in appendix.

### 3.2 Our New Algorithm

Each dual coordinate $y_i$ naturally corresponds to the $i$-th feature vector $a_i$. Therefore, given a raw clustering $[n] = S_1 \cup S_2 \cup \cdots \cup S_s$ of the dataset, we can partition the coordinates of the dual vector $y \in \mathbb{R}^n$ into $s$ blocks each corresponding to a cluster. Without loss of generality, we assume the coordinates of $y$ are sorted in the order of the cluster indices. In other words, we write $y = (y_{S_1}, \ldots, y_{S_s})$ where each $y_{S_c} \in \mathbb{R}^{n_c}$.

**Algorithm 1** ClusterACDM
---
**Input:** a raw clustering $S_1 \cup \cdots \cup S_s$.
 1: Apply cluster-based Haar transformation $H_{\mathsf{cl}}$ to get the transformed objective $D'(y')$.
 2: Run ACDM to minimize $D'(y')$
 3: Transform the solution of $D'(y')$ back to the original space.
---

ClusterACDM transforms the dual objective (3.2) into an equivalent form, by performing an $n_c$-dimensional Haar transformation on the $c$-th block of coordinates for every $c \in [s]$. Formally,

**Definition 3.2.** *Let* $R_2 \overset{\text{def}}{=} \begin{bmatrix} 1/\sqrt{2} & -1/\sqrt{2} \end{bmatrix}$, $R_3 \overset{\text{def}}{=} \begin{bmatrix} \sqrt{2}/\sqrt{3} & -\sqrt{2}/(2\sqrt{3}) & -\sqrt{2}/(2\sqrt{3}) \\ 0 & 1/\sqrt{2} & -1/\sqrt{2} \end{bmatrix}$,
*and more generally*

$$R_n \overset{\text{def}}{=} \begin{bmatrix} \frac{1/a}{\sqrt{1/a+1/b}} \cdots \frac{1/a}{\sqrt{1/a+1/b}} & \frac{-1/b}{\sqrt{1/a+1/b}} \cdots \frac{-1/b}{\sqrt{1/a+1/b}} \\ R_a & 0 \\ 0 & R_b \end{bmatrix} \in \mathbb{R}^{(n-1)\times n}$$

*for* $a = \lfloor n/2 \rfloor$ *and* $b = \lceil n/2 \rceil$. *Then, define the* $n$-*dimensional (**normalized**) Haar matrix as*

$$H_n \overset{\text{def}}{=} \begin{bmatrix} 1/\sqrt{n} \cdots 1/\sqrt{n} \\ R_n \end{bmatrix} \in \mathbb{R}^{n\times n}$$

We give a few examples of Haar matrices in Example A.2 in Appendix A. It is easy to verify that

**Lemma 3.3.** *For every* $n$, $H_n^T H_n = H_n H_n^T = I$, *so* $H_n$ *is a unitary matrix.*

**Definition 3.4.** *Given a clustering* $[n] = S_1 \cup \cdots \cup S_s$, *define the following **cluster-based Haar transformation** $H_{\mathsf{cl}} \in \mathbb{R}^{n \times n}$ that is a block diagonal matrix:*

$$H_{\mathsf{cl}} \overset{\text{def}}{=} \mathrm{diag}\big(H_{|S_1|}, H_{|S_2|}, \dots, H_{|S_s|}\big) \ .$$

*Accordingly, we apply the unitary transformation* $H_{\mathsf{cl}}$ *on (3.2) and consider*

$$\min_{y' \in \mathbb{R}^n} \left\{ D'(y') \overset{\text{def}}{=} \frac{1}{2n} \|y'\|^2 + \frac{1}{n} \langle y', H_{\mathsf{cl}} l \rangle + r^*\big( -\frac{1}{n} A H_{\mathsf{cl}}^T y' \big) \right\} \ . \tag{3.3}$$

*We call* $D'(y')$ *the **transformed objective** function.*

It is clear that the minimization problem (3.3) is equivalent to (3.2) by transforming $y = H_{\mathsf{cl}}^T y'$. Now, our ClusterACDM algorithm applies ACDM on minimizing this transformed objective $D'(y')$.

We claim the following running time of ClusterACDM and discuss the high-level intuition in the main body. We defer detailed analysis to Appendix A.

**Theorem 3.5.** *If* $r(\cdot)$ *is* $\lambda$-*strongly convex and an* $(s, \delta)$ *raw clustering is given, then ClusterACDM outputs an* $\varepsilon$-*approximate minimizer of* $D(\cdot)$ *in time* $T = O\big(nd + \frac{\max\{\sqrt{s}, \sqrt{\delta n}\}}{\sqrt{\lambda}} d\big)$ *.*

Comparing to the complexity of APCG, ACDM, or AccSDCA (see (1.1)), ClusterACDM is faster by a factor that is up to $\Omega\big(\min\{\sqrt{n/s}, \sqrt{1/\delta}\}\big)$.

**High-Level Intuition.** To see why Haar transformation is helpful, we focus on one cluster $c \in [s]$. Assume without loss of generality that cluster $c$ has vectors $a_1, a_2, \cdots, a_{n_c}$. After applying Haar transformation, the new columns $1, 2, \dots, n_c$ of matrix $A H_{\mathsf{cl}}^T$ become weighted combinations of $a_1, a_2, \cdots, a_{n_c}$, and the weights are determined by the entries in the corresponding rows of $H_{n_c}$.

Observe that every row except the first one in $H_{n_c}$ has its entries sum up to 0. Therefore, columns $2, \dots, n_c$ in $A H_{\mathsf{cl}}^T$ will be close to zero vectors and have small norms. In contrast, since the first row in $H_{n_c}$ has all its entries equal to $1/\sqrt{n_c}$, the first column of $A H_{\mathsf{cl}}^T$ becomes $\sqrt{n_c} \cdot \frac{a_1 + \cdots + a_{n_c}}{n_c}$, the scaled average of all vectors in this cluster. It has a large Euclidean norm.

The first column after Haar transformation can be viewed as an auxiliary feature vector representing the entire cluster. If we run ACDM with respect to this new matrix, and whenever this auxiliary column is selected, it represents "moving in the average direction of all vectors in this cluster". Since this single auxiliary column cannot represent the entire cluster, the remaining $n_c - 1$ columns serve as helpers that ensure that the algorithm is *unbiased* (i.e., converges to the exact minimizer).

Most importantly, as discussed in Remark 2.4, ACDM is a stochastic method that samples a dual coordinate $i$ (thus a primal feature vector $a_i$) with a probability proportional to its square-root

coordinate-smoothness (thus roughly proportional to $\|a_i\|$). Since auxiliary vectors have much larger Euclidean norms, we expect them to be sampled with probabilities much larger $1/n$. This is how the faster running time is obtained in Theorem 3.5.

REMARK. The speed-up of ClusterACDM depends on how much "non-uniformity" the underlying coordinate descent method can utilize. Therefore, no speed-up can be obtained if one applies APCG instead of the NU_ACDM which is optimally designed to utilize coordinate non-uniformity.

## 4 ClusterSVRG Algorithm

Our ClusterSVRG is a *non-accelerated* stochastic gradient method just like SVRG [13], SAGA [6], SDCA [22], etc. It directly works on minimizing the primal objective (similar to SVRG and SAGA):

$$\min_{x \in \mathbb{R}^d} \left\{ F(x) \overset{\text{def}}{=} f(x) + \Psi(x) \overset{\text{def}}{=} \frac{1}{n} \sum_{i=1}^{n} f_i(x) + \Psi(x) \right\} . \tag{4.1}$$

Here, $f(x) = \frac{1}{n} \sum_{i=1}^{n} f_i(x)$ is the finite average of $n$ functions, each $f_i(x)$ is convex and $L$-smooth, and $\Psi(x)$ is a simple (but possibly non-differentiable) convex function, sometimes called the proximal function. We denote $x^*$ as a minimizer of (4.1).

Recall that stochastic gradient methods work as follows. At every iteration $t$, they perform updates $x^t \leftarrow x^{t-1} - \eta \widetilde{\nabla}^{t-1}$ for some step length $\eta > 0$,[6] where $\widetilde{\nabla}^{t-1}$ is the so-called *gradient estimator* and its expectation had better equal the full gradient $\nabla f(x^{t-1})$. It is a known fact that the faster the variance $\mathbf{Var}[\widetilde{\nabla}^{t-1}]$ diminishes, the faster the underlying method converges. [13]

For instance, SVRG defines the estimator as follows. It has an outer loop of epochs. At the beginning of each epoch, SVRG records the current iterate $x$ as a snapshot point $\widetilde{x}$, and computes its full gradient $\nabla f(\widetilde{x})$. In each inner iteration within an epoch, SVRG defines $\widetilde{\nabla}^{t-1} \overset{\text{def}}{=} \frac{1}{n} \sum_{j=1}^{n} \nabla f_j(\widetilde{x}) + \nabla f_i(x^{t-1}) - \nabla f_i(\widetilde{x})$ where $i$ is a random index in $[n]$. SVRG usually chooses the epoch length $m$ to be $2n$, and it is known that $\mathbf{Var}[\widetilde{\nabla}^{t-1}]$ approaches to zero as $t$ increases. We denote by $\widetilde{\nabla}^{t-1}_{\text{SVRG}}$ this choice of $\widetilde{\nabla}^{t-1}$ for SVRG.

In ClusterSVRG, we define the gradient estimator $\widetilde{\nabla}^{t-1}$ based on clustering information. Given a clustering $[n] = S_1 \cup \cdots \cup S_s$ and denoting by $c_i \in [s]$ the cluster that index $i$ belongs to, we define

$$\widetilde{\nabla}^{t-1} \overset{\text{def}}{=} \frac{1}{n} \sum_{j=1}^{n} \left( \nabla f_j(\widetilde{x}) + \zeta_{c_j} \right) + \nabla f_i(x^{t-1}) - \left( \nabla f_i(\widetilde{x}) + \zeta_{c_i} \right) .$$

Above, for each cluster $c$ we introduce an additional $\zeta_c$ term that can be defined in one of the following two ways. Initializing $\zeta_c = 0$ at the beginning of the epoch for each cluster $c$. Then,

- In Option I, after each iteration $t$ is completed and suppose $i$ is the random index chosen at iteration $t$, we update $\zeta_{c_i} \leftarrow \nabla f_i(x^{t-1}) - \nabla f_i(\widetilde{x})$.
- In Option II, we divide an epoch into subepochs of length $s$ each (recall $s$ is the number of clusters). At the beginning of each subepoch, for each cluster $c \in [s]$, we define $\zeta_c \leftarrow \nabla f_j(\overline{x}) - \nabla f_j(\widetilde{x})$. Here, $\overline{x}$ is the last iterate of the previous subepoch and $j$ is a random index in $S_c$.

We summarize both options in Algorithm 2. Note that Option I gives simpler intuition but Option II leads to a simpler proof.

The intuition behind our new choice of $\widetilde{\nabla}^{t-1}$ can be understood as follows. Observe that in the SVRG estimator $\widetilde{\nabla}^{t-1}_{\text{SVRG}}$, each term $\nabla f_j(\widetilde{x})$ can be viewed as a *"guess term"* of the true gradient $\nabla f_j(x^{t-1})$ for function $f_j$. However, these guess terms may be very "outdated" because $\widetilde{x}$ can be $m = 2n$ iterations away from $x^{t-1}$, and therefore contribute to a large variance.

We use raw clusterings to improve these guess terms and reduce the variance. If function $f_j$ belongs to cluster $c$, then our Option I uses $\nabla f_j(\widetilde{x}) + \nabla f_k(x^t) - \nabla f_k(\widetilde{x})$ as the *new* guess of $\nabla f_j(x^t)$, where $t$ is the last time cluster $c$ was accessed and $k$ is the index of the vector in this cluster that was accessed. This new guess only has an "outdatedness" of roughly $s$ that could be much smaller than $n$.

Due to space limitation, we defer all technical details of ClusterSVRG to Appendix B and B.3.

**SVRG vs. SAGA vs. ClusterSVRG.** SVRG becomes a special case of ClusterSVRG when all the data vectors belong to the *same* cluster; SAGA becomes a special case of ClusterSVRG when each

**Algorithm 2** ClusterSVRG

---

**Input:** Epoch length $m$ and learning rate $\eta$, a raw clustering $S_1 \cup \cdots \cup S_s$.
1: $x^0, \overline{x} \leftarrow$ initial point, $t \leftarrow 0$.
2: **for** epoch $\leftarrow 0$ **to** MaxEpoch **do**
3:      $\widetilde{x} \leftarrow x^t$, and $(\zeta_1, \ldots, \zeta_s) \leftarrow (0, \ldots, 0)$
4:      **for** iter $\leftarrow 1$ **to** $m$ **do**
5:          $t \leftarrow t+1$ and choose $i$ uniformly at random from $\{1, \cdots, n\}$
6:          $x^t \leftarrow x^{t-1} - \eta \left( \frac{1}{n} \sum_{j=1}^n \left( \nabla f_j(\widetilde{x}) + \zeta_{c_j} \right) + \nabla f_i(x^{t-1}) - \left( \nabla f_i(\widetilde{x}) + \zeta_{c_i} \right) \right)$
7:          **Option I:** $\zeta_{c_i} \leftarrow \nabla f_i(x^{t-1}) - \nabla f_i(\widetilde{x})$
8:          **Option II: if** $iter$ **mod** $s = 0$ **then** for all $c = 1, \ldots, s$,
9:             $\zeta_c \leftarrow \nabla f_j(x^{t-1}) - \nabla f_j(\widetilde{x})$ where $j$ is randomly chosen from $S_c$.
10:      **end for**
11: **end for**

---

data vector belongs to its *own* cluster. We hope that this interpolation helps experimentalists decide between these methods: (1) if the data vectors are pairwise close to each other then use SVRG; (2) if the data vectors are all very separated from each other then use SAGA; and (3) if the data vectors have nice clustering structures (which one can detect using LSH), then use our ClusterSVRG.

## 5 Experiments

We conduct experiments for three datasets that can be found on the LibSVM website [8]: COV-TYPE.BINARY, SENSIT (combined scale), and NEWS20.BINARY. To make easier comparison across datasets, we scale every vector by the average Euclidean norm of all the vectors. This step is for comparison only and not necessary in practice. Note that Covtype and SensIT are two datasets where the feature vectors have a nice clustering structure; in contrast, dataset News20 cannot be well clustered and we include it for comparison purpose only.

### 5.1 Clustering and Haar Transformation

We use the approximate nearest neighbor algorithm library E2LSH [2] to compute raw clusterings. Since this is not the main focus of our paper, we include our implementation in Appendix D. The running time needed for raw clustering is reasonable. In Table 1 in the appendix, we list the running time (1) to sub-sample and detect if good clustering exists and (2) to compute the actual clustering. We also list the one-pass running time of SAGA using sparse implementation for comparison.

We conclude two things from Table 1. First, in about the same time as SAGA performing $0.3$ pass on the datasets, we can detect clustering structure in the dataset for a given diameter $\delta$. This is a fast-enough preprocessing step to help experimentalists choose to use clustering-based methods or not. Second, in about the same time as SAGA performing 3 passes on well-clustered datasets such as Covtype and SensIT, we obtain the actual raw clustering. As emphasized in the introduction, we view the time needed for clustering as *negligible*. This not only because $0.3$ and $3$ are small quantities as compared to the average number of passes needed to converge (which is usually around 20). It is also because the clustering time is usually amortized over multiple runs of the training algorithm due to different data analysis tasks, parameter tunings, etc.

In ClusterACDM, we need to pre-compute matrix $AH_{\text{cl}}^T$ using Haar transformation. This can be efficiently implemented thanks to the sparsity of Haar matrices. In Table 2 in the appendix, we see that the time needed to do so is roughly 2 passes of the dataset. Again, this time should be amortized over multiple runs of the algorithm so is negligible.

### 5.2 Performance Comparison

We compare our algorithms with SVRG, SAGA and ACDM. We use default epoch length $m = 2n$ and Option I for SVRG. We use $m = 2n$ and Option I for ClusterSVRG. We consider ridge and Lasso regressions, and denote by $\lambda$ the weight of the $\ell_2$ regularizer for ridge or the $\ell_1$ regularizer for Lasso.

**Parameters.** For SVRG and SAGA, we tune the best step size for each test case. To make our comparison even stronger, instead of tuning the best step size for ClusterSVRG, we simply set it to be either the best of SVRG or the best of SAGA in each test case. For ACDM and ClusterACDM, the step size is computed automatically so tuning is unnecessary.

For Lasso, because the objective is not strongly convex, one has to add a dummy $\ell_2$ regularizer on the objective in order to run ACDM or ClusterACDM. (This step is needed for every accelerated method

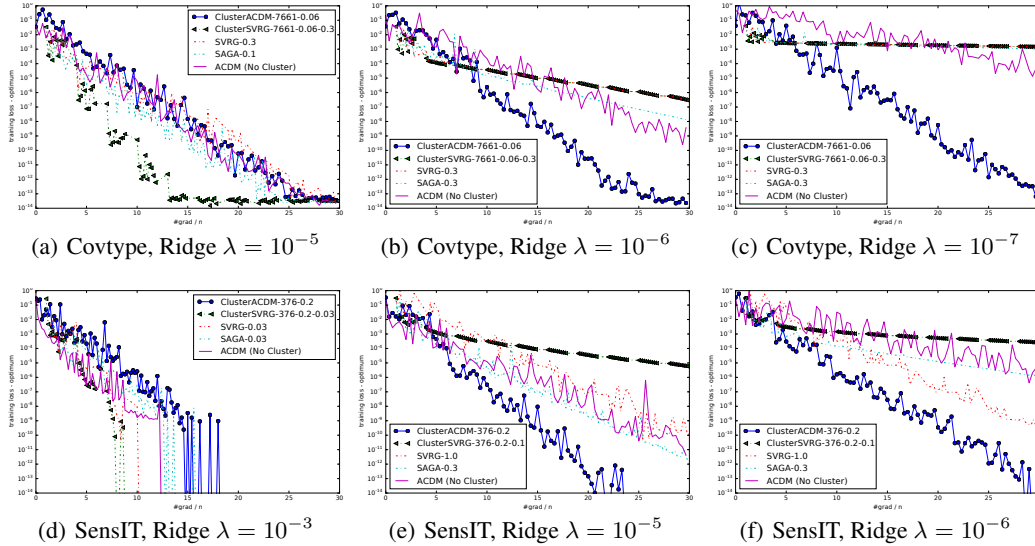

(a) Covtype, Ridge $\lambda = 10^{-5}$     (b) Covtype, Ridge $\lambda = 10^{-6}$     (c) Covtype, Ridge $\lambda = 10^{-7}$

(d) SensIT, Ridge $\lambda = 10^{-3}$     (e) SensIT, Ridge $\lambda = 10^{-5}$     (f) SensIT, Ridge $\lambda = 10^{-6}$

Figure 1: Selected plots on ridge regression. For Lasso and more detailed comparisons, see Appendix

including AccSDCA, APCG or SPDC.) We choose this dummy regularizer to have weight $10^{-7}$ for Covtype and SenseIT, and weight $10^{-6}$ for News20.[7]

**Plot Format.** In our plots, the $y$-axis represents the objective distance to the minimizer, and the $x$-axis represents the number of passes of the dataset. (The snapshot computation of SVRG and ClusterSVRG counts as one pass.) In the legend, we use the format

- "ClusterSVRG–$s$–$\delta$–stepsize" for ClusterSVRG,
- "ClusterACDM–$s$–$\delta$" for ClusterACDM.
- "SVRG/SAGA–stepsize" for SVRG or SAGA.
- "ACDM (no Cluster)" for the vanilla ACDM without using any clustering info.[8]

**Results.** Our comprehensive experimental plots are included only in the appendix, see Figure 2, 3, 4, 5), 6, and 7. Due to space limitation, here we simply compare all the algorithms on ridge regression for datasets SensIT and Covtype by choosing only one representative clustering, see Figure 1.

Generally, ClusterSVRG outperforms SAGA/SVRG when the regularizing parameter $\lambda$ is *large*. ClusterACDM outperforms all other algorithms when $\lambda$ is *small*. This is because accelerated methods outperform non-accelerated ones with smaller values of $\lambda$, and the complexity of ClusterACDM outperforms ACDM more when $\lambda$ is smaller (compare (1.1) with (1.2)).[9]

Our other findings can be summarized as follows. Firstly, dataset News20 does not have a nice clustering structure but our ClusterSVRG and ClusterACDM still perform comparably well to SVRG and ACDM respectively. Secondly, the performance of ClusterSVRG is slightly better with clustering that has smaller diameter $\delta$. In contrast, ClusterACDM with larger $\delta$ performs slightly better. This is because ClusterACDM can take advantage of very large but low-quality clusters, and this is a very appealing feature in practice.

**Sensitivity on Clustering.** In Figure 8 in appendix, we plot the performance curves of ClusterSVRG and ClusterACDM for SensIT and Covtype, with 7 different clusterings. From the plots we claim that ClusterSVRG and ClusterACDM are very insensitive to the clustering quality. As long as one does not choose the most extreme clustering, the performance improvement due to clustering can be significant. Moreover, ClusterSVRG is slightly faster if the clustering has relatively smaller diameter $\delta$ (say, below 0.1), while the ClusterACDM can be fast even for very large $\delta$ (say, around 0.6).

## Footnotes

[3]N-SAGA uses the stochastic gradient of one data vector to completely represent its neighbors. This changes the objective value and therefore cannot give very accurate solutions.

[4]The asymptotic worst-case running time for non-accelerated methods in (1.1) cannot be improved in general, even if a perfect clustering (i.e., $\delta = 0$) is given.

[5]They showed that defining $x = \nabla r^*(-Ay/n)$, if $y$ is a good approximate minimizer of the dual objective $D(y)$, $x$ is also a good approximate minimizer of the primal objective $P(x)$.

[6] Or more generally the proximal updates $x^t \leftarrow \arg\min_x \left\{ \frac{1}{2\eta} \|x - x^{t-1}\|^2 + \langle \widetilde{\nabla}^{t-1}, x \rangle + \Psi(x) \right\}$ if $\Psi(x)$ is nonzero.

[7]Choosing large dummy regularizer makes the algorithm converge faster but to a worse minimum, and vice versa. In our experiments, we find these choices reasonable for our datasets. Since our main focus is to compare ClusterACDM with ACDM, as long as we choose the same dummy regularizer our the comparison is *fair*.

[8]ACDM has slightly better performance compared to APCG, so we adopt ACDM in our experiments [1]. Furthermore, our comparison is fair because ClusterACDM and ACDM are implemented in the same manner.

[9]The best choice of $\lambda$ usually requires cross-validation. For instance, by performing a 10-fold cross validation, one can figure out that the best $\lambda$ is around $10^{-6}$ for SensIT Ridge, $10^{-5}$ for SensIT Lasso, $10^{-7}$ for Covtype Ridge, and $10^{-6}$ for Covtype Lasso. Therefore, for these two datasets ClusterACDM is preferred.

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
