[Reviews · NeurIPS 2016]

Reviewer 1

Summary

The paper introduces two new algorithms: cluster version of accelerated coordinate descent methods (Cluster ACDM) and cluster version of stochastic variance reduced gradient method (Cluster SVRG). The basic idea is to study how it is possible to improve these methods using additional information on the data; in particular: clustering information. The authors define several notions of raw clusters (different for the two methods; but related) – and explain how these can be efficiently computed. Using the clustering information, the SVRG and ACDM methods are then accelerated. The resulting complexity results are better than the results for the previous variants – as they do not make use of such extra closeting information. The theoretical findings are sufficiently conclusively confirmed by experiments; with insightful commentary. For ACDM, the new approach can be seen as a new preconditioner enhanced by a novel Haar transform (specific rotations on each cluster highlighting average direction in each cluster and suppressing individual directions in a stochastic sense); and for SVRG, the the new Cluster SVRG method can be seen as an algorithm interpolating between SVRG and SAGA; with further reduction of variance of the stochastic gradient enabled by reduction in “outdatedness” via utilizing clustering information. In general, this is a beautifully written paper, with elegant new results in an important area of ML.

Qualitative Assessment

The initial motivation seems to be the work of Hoffman et al on the use of clustering to speedup stochastic methods for ERM. Their method was not proved to converge to the optimal due to the use of biased stochastic gradients. Also, that work seemed to work only for small clusters due to the approach chosen. This papers goes a long way to develop the basic idea into a satisfying theoretical framework which also gives rise to efficient implementations. This paper is truly a pleasure to read – a very fine example of academic exposition. The results are innovative, and yet intuitively explained. The theory seems correct; and the experiments show that the methods work better than the competing approaches considered. I believe that several people might pick up this work in an attempt to use it / extend the results. Question 1: Can Assumption B.3 be removed or replaced by something that can be verified from assumptions on the problem only (and not on the algorithm itself)? Can you prove that in some cases \xi is finite? What value does it have? How large can it be? Question 2: What are the challenges associated with extending the Cluster ACDM results to non-quadratic losses? Question 3: Is Haar transform necessary? What other transforms can have the same effect on the dual problem? Small issues: 1. Line 60: improve -> expect 2. Line 125: can be -> to be 3. Line 178: H_{cl} is not from R^n 4. Lines 263 and 266: same as -> same time as 5. Line 296: ride -> ridge

Confidence in this Review

3-Expert (read the paper in detail, know the area, quite certain of my opinion)


Reviewer 2

Summary

The paper exploits the idea of clustering data prior to performing ERM with a convex (regularized) objective. Very cleverly, the authors perform a variable change in the dual, using a Harr transform, which associates the "cluster average" to a single variable and the deviation from the cluster centroid to other dimensions. They then exploit non-uniform sampling to improve upon a non-clustered version of stochastic dual coordinate ascent.

Qualitative Assessment

I really like the key idea of the paper. It is very convincing and clean. To exploit the clustering idea in the dual space via Haar-transform is a highly original and effective insight. It allows the authors to leverage a lot of recent work on accelerating standard SGD. The main weakness in my view is the writeup. It needs more work and polishing when it comes to language and explanations. However, I feel that with a solid idea and analysis, this can easily be addressed in the final version of the paper.

Confidence in this Review

3-Expert (read the paper in detail, know the area, quite certain of my opinion)


Reviewer 3

Summary

By introducing the notion of raw clusters, the authors propose two algorithms to minimize an empirical risk function. The first one is the Cluster ACDM. This algorithm is an application of ACDM to the dual problem after doing a change of variable defined by Haar transformation (see (3.3)). The complexity is obtained when the regularized function is strongly convex. The second one is the ClusterSVRG algorithm. In this algorithm, based on cluster information, a new stochastic estimate of the gradient is introduced. The resulting algorithm is the interpolation between SVRG and SAGA that are two extreme cases. Comparisons and numerical experiments are also provided.

Qualitative Assessment

This work can be viewed to address the practical issues of ACDM and SVRG, SAGA.

Confidence in this Review

1-Less confident (might not have understood significant parts)


Reviewer 4

Summary

In this paper authors introduced ClusterACDM and ClusterSVRG methods for the empirical risk minimization. For clustering data points, instead of exact clustering which is computationally expensive, author use Raw Clustering algorithms such as LSH that quickly converge to a reasonably good clustering. ClusterACDM uses the clustering in the dual space, coupled with inexpensive Haar Transformation, which is introduced to remove the bias, to provide a faster convergence than ACDM. ClusterSVRG which is a generalized method created on top of SVRG algorithm that contains N-SAGA and SVRG as its special cases, provides a faster theoretical guarantees than existing non-accelerated and accelerated methods. The simulations show that these methods work better than all existing ones over some datasets.

Qualitative Assessment

This paper introduces two novel methods for the empirical risk minimization problem. The authors have done a great job on studying the literature and explaining how to efficiently take into account the clustering aspects of data. Figures and tables are well-representative of the result. ClusterACDM and ClusterSVRG methods both are really novel and at the same time general and they perform really well in practice. I believe that this paper is a strong submission. I have a few minor comments/questions that are listed below: 1) ClusterACDM and ClusterSVRG need to parameters to be specified $s$ and $\delta$. Is there any possible way to choose the best possible pair or instead we need to tune them manually ? What about the simulations on paper ? For ClusterSVRG, based on the upper bound on the worst-case running time, these parameters had better to be related by $s= \delta n$, but is it possible to achieve this ? 2) In figure 8, I would recommend using $10^{-2}$ instead of $1e-2$ which is not consistent to previous representations. To conclude, I believe the authors have presented their proposed method and its performance well and I consider this submission as an instructive one.

Confidence in this Review

2-Confident (read it all; understood it all reasonably well)


Reviewer 5

Summary

This paper proposes to use raw clusters to speed up optimization algorithms. Raw clusters group similar data together, and the proposed algorithms would use clustering information to speed up the optimization process. Both dual-based algorithm (ClusterACDM) and primal-based algorithm (ClusterSVRG) are proposed. Convergence analysis and numerical results are also provided.

Qualitative Assessment

This paper is well written and to the best of my knowledge, the main ideas of this paper (Haar transformation and new gradient estimator using clustering) are novel. The convergence results and numerical results show that the proposed algorithms are promising. I still have minor concerns about the sensitivity of the cluster choice (s,\delta) and the complexity of using clustering algorithms even though they are addressed in the paper. Is there any theoretical result of the complexity of using clustering algorithms? How does it scale with n and d? What is the worse case it would be if the input clusters are not correct/useful at all? == post-rebuttal update== I have read the authors' rebuttal, and I think extra information on existing clustering algorithms would be enlightening. Whether these existing clustering algorithms works well or not, it does not discount the contributions of this paper.

Confidence in this Review

2-Confident (read it all; understood it all reasonably well)


Reviewer 6

Summary

The authors present two new algorithms ClusterACDM and ClusterSVRG, which share the common idea of using a clustering of the data vectors to find a better trade-off between speed and accuracy of stochastic gradient descend methods. In the ClusterACDM case they consider the dual of the regularized least-square problem and transform it with a cluster-based block diagonal Haar transformation, to which the ACDM is applied. This unitary transformation on the input data generates a single representative for each cluster and the other vectors in each cluster are almost zero (if the vectors within one cluster were close to each other at the beginning). The ClusterSVRG algorithm minimizes the average of $n$ convex L-smooth functions $f_i$ plus a possibly non-differentiable convex function. The algorithm is based on SVRG which is divided into epochs, while at the beginning of each epoch the gradient (at that point) is calculated exactly. Within one epoch at each step a variation of this gradient is used, i.e., one function is chosen and the contribution of this function to the gradient is exchanged with the gradient at the new point. The novelty this paper introduces is that for each cluster of the data (in this case the convex functions $f_i$) the last gradient of a member of this cluster is used as an approximation of the gradient for all members of this cluster instead of the exact computation at the beginning of the epoch.

Qualitative Assessment

It is also not clear how to run the detection phase of the raw clustering algorithm, since the time window to run the algorithm is given relative to the time SAGA takes to run. So how are the algorithms executed when there is no implementation of SAGA available? It is good that the appendix has some more experiments, however some explanation of the graphs would be good. Especially since in many cases the proposed algorithms do not perform better than the benchmark algorithms.

Confidence in this Review

2-Confident (read it all; understood it all reasonably well)